# VOC 202012/01 Variant Is Effectively Neutralized by Antibodies Produced by Patients Infected before Its Diffusion in Italy

**DOI:** 10.3390/v13020276

**Published:** 2021-02-11

**Authors:** Valeria Rondinone, Lorenzo Pace, Antonio Fasanella, Viviana Manzulli, Antonio Parisi, Maria Rosaria Capobianchi, Angelo Ostuni, Maria Chironna, Elisabetta Caprioli, Maria Labonia, Dora Cipolletta, Ines Della Rovere, Luigina Serrecchia, Fiorenza Petruzzi, Germana Pennuzzi, Domenico Galante

**Affiliations:** 1Istituto Zooprofilattico Sperimentale of Puglia and Basilicata, 71121 Foggia, Italy; valeria.rondinone@izspb.it (V.R.); antonio.fasanella@izspb.it (A.F.); viviana.manzulli@izspb.it (V.M.); antonio.parisi@izspb.it (A.P.); dora.cipolletta@izspb.it (D.C.); inesdellarovere@iizspb.it (I.D.R.); luigina.serrecchia@izspb.it (L.S.); fiorenza.petruzzi@izspb.it (F.P.); germana.pennuzzi@izspb.it (G.P.); domenico.galante@izspb.it (D.G.); 2Laboratory of Virology, National Institute for Infectious Diseases “L. Spallanzani”, 00149 Roma, Italy; maria.capobianchi@inmi.it; 3Transfusional Medicine Unit, University-Hospital Policlinico, 70124 Bari, Italy; angelo.ostuni@policlinico.ba.it; 4Department of Biomedical Sciences and Human Oncology-Hygiene Section, University of Bari “Aldo Moro”, 70124 Bari, Italy; maria.chironna@uniba.it; 5Clinical Pathology Laboratory, “Di Miccoli” Hospital, 70051 Barletta, Italy; elisabetta.caprioli@aslbat.it; 6Microbiology Laboratory, IRCCS Casa Sollievo della Sofferenza, 71013 San Giovanni Rotondo, Italy; m.labonia@operapadrepio.it

**Keywords:** Covid-19, neutralizing antibodies, SARS-CoV-2 VOC 202012/01 variant

## Abstract

The coronavirus disease 2019 (Covid-19) pandemic is caused by the severe acute respiratory syndrome coronavirus 2 (SARS-CoV-2) and presents a global health emergency that needs urgent intervention. Viruses constantly change through mutation, and new variants of a virus are expected to occur over time. In the United Kingdom (UK), a new variant called B.1.1.7 has emerged with an unusually large number of mutations. The aim of this study is to evaluate the level of protection of sera from 12 patients infected and later healed in Apulia Region (Italy) with Covid-19 between March and November 2020, when the English variant was not circulating in this territory yet, against the new VOC 202012/01 variant by seroneutralization assay. The sera of patients had already been tested before, using a virus belonging to the lineage B.1 and showed an antibody neutralizing titer ranging between 1:160 and 1:320. All the 12 sera donors confirmed the same titers of neutralizing antibodies obtained with a strain belonging to the lineage B.1.1.7 (VOC 202012/01). These data indicate that antibodies produced in subjects infected with variants of Sars-CoV-2 strain before the appearance of the English one, seem to have a neutralizing power also against this variant.

## 1. Introduction

Viruses, in particular RNA viruses, are often subject to mutations, and the appearance of new variants is a quite a common event. The SARS-CoV-2 variant virus that has emerged in the UK in December 2020, the so-called “English variant” (VOC 202012/01), attracts particular attention due to its high contagiousness [1]. This variant presents multiple spike protein mutations (deletion 69–70, deletion 144, substitutions N501Y, A570D, D614G, P681H, T716I, S982A, and D1118H) as well as mutations in other genomic regions [2]. Among them, N501Y is of major concern because it involves one of the six key amino acid residues determining a tight interaction of the SARS-CoV-2 receptor-binding domain (RBD) with its cellular receptor angiotensin-converting enzyme 2 (ACE2) [3,4]. This mutation has also appeared, independently, in a rapidly spreading variant in South Africa [5].

Preliminary reports by the United Kingdom indicate that this variant is more transmissible than previous circulating strains, with an estimated increase of between 40% and 70% in transmissibility [6].

Laboratory studies are ongoing to determine whether these variant viruses have different biological properties or may alter vaccine efficacy [7].

Currently, there is no evidence that these variants cause more severe illness or increased risk of death. On the other hand, the risk that substantial genomic alteration may lead to test failure or escape from antibody response or vaccine efficacy must be carefully considered, and new information about the virologic, epidemiologic, and clinical characteristics of this variants is rapidly emerging [8]. As a matter of fact, a better understanding of how easily these variants might be transmitted and whether currently authorized vaccines will protect people against them is needed.

In addition, convalescent plasma (hyperimmune plasma) is one of the most used therapeutic strategies to reduce mortality and severity of the disease in patients infected with Covid-19 [9], but there are still no data about the efficacy of hyperimmune plasma obtained by patients infected and then healed before December 2020 against this variant that is rapidly spreading across Europe.

The aim of this study is to demonstrate that neutralizing antibodies produced by Italian patients who acquired the infection from March to November 2020, before the spread of VOC 202112/01, are fully protective also against this variant.

## 2. Materials and Methods

### 2.1. Patients

Serum samples collected from 12 patients who acquired the Covid-19 infection between March 2020 and November 2020, were recruited as hyperimmune plasma donors for the Italian project TSUNAMI and used in the present study. All serum samples had been previously analysed by seroneutralization test using a strain isolated in Italy during the first stage of the pandemic, and all of them showed an antibody neutralizing titer ranging between 1:160 and 1:320.

The lineage of the infecting virus was established for two patients, for whom the original diagnostic sample collected at the time of diagnosis could be retrospectively analysed by whole genome sequencing. For the other patients, for whom the infecting viral strain could not be retrospectively established, it is reasonable to assume that the strains were different from VOC 202012/01, since during the period in which the patients became infected, this variant did not circulate in the Apulia Region.

The seroneutralization test was run in parallel with the VOC 202012/01 strain isolated in a biosafety level 3 (BSL-3) laboratory of the Istituto Zooprofilattico Sperimentale of Puglia and Basilicata (Foggia, Italy), and with the virus isolate that has been used for the standardization of the test protocol that was established within the Italian network for SARS-CoV-2 seroneutralization test (NeuCoV-Net) established for the TSUNAMI protocol. The first one (GISAID accession number: EPI_ISL_745193), belonging to the lineage B.1.1.7 virus (VOC 202012/01), was isolated from a patient returning from Great Britain on 22nd of December 2020; the second one, belonging to the lineage B.1 clade G (GISAID accession number: EPI_ISL_568579), was isolated in March 2020 in Italy and was provided by Prof. Fausto Baldanti, Fondazione IRCCS Policlinico San Matteo (Pavia, Italy).

### 2.2. Cells and Virus Stock

African green monkey kidney Vero E6 cells were used for both the propagation of SARS-CoV-2 and the neutralization assay. Cells were cultured into a 25 cm^2^ cell culture flask in Eagle’s minimal essential medium (EMEM) (Life Technologies, Carisbad, CA, USA) supplemented with 10% (*v*/*v*) fetal bovine serum (FBS) (Life Technologies, Carisbad, CA, USA), and 100 U/mL penicillin and streptomycin (Life Technologies, Carisbad, CA, USA) in 5% CO_2_ at 37 °C.

All the procedures handling the SARS-CoV-2 and infected cell cultures were held in a BSL-3 laboratory. To produce the viral stock, the virus was propagated in Vero E6 cells, and culture medium was collected and centrifugate to remove the cell debris [10].

### 2.3. Titration of SARS-CoV-2

Virus infectious titers were established by the Reed and Muench tissue culture infective dose (TCID_50_) end point method [11]. For titration, 2 × 10^4^ Vero E6 cells (in 50 µL) were plated into 96-well plates. Then the stock solution of SARS-CoV-2 was diluted serially from 10^−1^ to 10^−8^, and 25 µL of each dilution was added to the cells and incubated in 5% CO_2_ at 37 °C for 72 h. Eight replicates were performed for each dilution and used to quantify the virus titer and statistically determine the TCID_50_ end point.

### 2.4. Cytopathic Effect Based Micro-Neutralization Assay

A cytopathic effect (CPE)-based micro-neutralization assay was conducted in 96-well microtiter plates [12]. Briefly, eight-fold dilutions (from 1:10 to 1:640) of human serum samples were tested in triplicate wells for the presence of antibodies that neutralized the infectivity of SARS-Cov-2 in Vero E6 cell monolayers. 100 TCID_50_ of virus in 25 µL/well were incubated with 25 µL of each dilution of serum in EMEM with 6% FBS for 1 h at 37 °C. After the incubation, 2 × 10^4^ Vero E6 cells (in 50 µL) were added to each well. The results of the seroneutralization test were determined by the appearance after 72 h of CPE, which was observed under an inverted microscope Axiovert 25 (Zeiss, Oberkocken, Germany). The neutralizing antibody titer was defined as the highest serum dilution at which no CPE breakthrough in any of the testing wells was observed.

## 3. Results

Table 1 shows the comparison of seroneutralization titer of serum samples from 12 donors, who had been infected with SARS-COV-2 strains different from VOC 202012/01, assessed against two different SARS-CoV-2 strains: i. lineage B.1 and ii. lineage B.1.1.7 (VOC 202012/01). As can be seen, the seroneutralization titer was identical against both viral strains (Table 1). Table 2 highlights the main mutations harbored by B.1 and B.1.1.7 lineages.

## 4. Discussion

The occurrence of new variants in the evolution of Sars-CoV-2 is a natural event, but it raises concern as mutations may determine altered biologic characteristics of the virus, including increased contagiousness, as it is the case of VOC 202012/01, which appeared in the UK and spread to several countries. In fact, this variant harbors multiple mutations in the spike protein, improving the ability of the virus to bind and penetrate cells. One major concern is the possibility that a profoundly mutated spike protein may alter the recognition by neutralizing antibodies, raising concerns about the protective ability of the recently issued vaccines that are based on original spike protein sequences. The evidence provided in the present study, according to a recent report [13], indicates that the VOC202012/01 variant is sensitive to the neutralizing activity of antibodies produced by patients in response to previously circulating viral strains, and the neutralizing titers are identical to those established by using as virus challenge a different strain that was isolated several months ago and is used nationwide to establish the neutralization titer of hyperimmune convalescent plasma preparations in Italy. These data are reassuring, in that all those who have overcome the disease and have produced good levels of protective antibodies may be protected against a possible reinfection sustained by an even distant SARS-CoV-2 strain. Furthermore, the administration of hyperimmune plasma may be considered a good protective opportunity against distant strains of SARS-CoV-2, irrespectively of the characteristics of the strains that had induced the antibody response. In line with these considerations, Xuping Xie et al. [14] have recently shown that antibodies produced by the mRNA-based COVID-19 vaccine BNT162b2 are effective against the isogenic Y501 SARS-CoV-2 developed on the genetic background of the N501 clinical strain USA- WA1/2020, which also provided the genetic background of the BNT162b2-encoded spike antigen. With respect to this study, our evidence adds important information, as the viruses used in the present study are authentic circulating strains that are both sensitive to the same extent to antibodies elicited by different viral strains.

## 5. Conclusions

The data of this study indicate that the antibodies produced in subjects infected with variants of Sars-CoV-2 strain circulating before the appearance of the VOC 202012/01 variant possess the same neutralizing power against this variant. Genomic surveillance aimed at performing an updated molecular map of circulating Sars-CoV-2 strains plays a fundamental role, since this virus can easily undergo genetic variations. We cannot exclude a priori that virus mutation(s) in the future might induce such a relevant structural change of the spike protein to require vaccine update and modification.

## Figures and Tables

**Table 1 viruses-13-00276-t001:** Data of the analysed patients in this study and results of seroneutralization tests. The symptoms refer to the date of the first positive test. n.a.d. (not available data).

*n*	Age	Date of Positivity (Molecular Test)	Symptoms	Genotype of Sars-CoV-2 Strain Isolated from the Patient	Genotype of Sars-CoV-2 Strain Predominating in the Territory	Date of Serum Collection	Seroneutralization titer against Lineage B.1	Seroneutralization Titer against Lineage B.1.1.7	
1	40	10 March 2020	fever, cough	n.a.d.	B.1	8 October 2020	1:160	1:160
2	45	12 March 2020	fever, cough,	n.a.d.	B.1	8 October 2020	1:160	1:160
			difficulty in breathing					
3	60	21 March 2020	asymptomatic	n.a.d.	B.1	17 September 2020	1:160	1:160
4	60	4 April 2020	fever	n.a.d.	B.1	17 September 2020	1:160	1:160
5	23	18 October 2020	fever	B.1.177	B.1.177	2 November 2020	1:160	1:160
6	52	26 October 2020	fever, cough	B.1.1.229	B.1.177	7 December 2020	1:160	1:160
7	59	27 October 2020	fever, cough,	n.a.d.	B.1.177	23 December 2020	1:320	1:320
			difficulty in breathing					
8	49	31 October 2020	fever, cough	n.a.d.	B.1.177	21 December 2020	1:160	1:160
9	25	31 October 2020	fever	n.a.d.	B.1.177	17 December 2020	1:160	1:160
10	53	17 November 2020	fever, cough	n.a.d.	B.1.177	23 December 2020	1:320	1:320
11	35	21 November 2020	fever, cough	n.a.d.	B.1.177	23 December 2020	1:320	1:320
12	27	26 November 2020	fever, cough	n.a.d.	B.1.177	23 December 2020	1:320	1:320

**Table 2 viruses-13-00276-t002:** Main mutations characterizing B.1 and B.1.1.7 lineages vs. the reference strain NC_04551.2.

	B.1.1.7		B.1	
Gene	Nucleotide	Amino Acid	Nucleotide	Amino Acid
ORF1ab	C3267T	T1001I	A3569G	S1102G
	C5388A	A1708D	C14408T	P314L
	T6954C	I2230T		
	11288-11296 del.	SGF 3675-3677 del.		
S	21765-21770 del.	HV 69-70 del.	G22205C	D215H
	21991-21993 del.	Y144 del.	A23403G	D614G
	A23063T	N501Y	C23481T	S640F
	C23271A	A570D	C23525T	H655Y
	C23604A	P681H		
	C23709T	T716I		
	T24506G	S982A		
	G24914C	D1118H		
N	28280 GAT->CTA	D3L		
	C28977T	S235F		
E			A26369G	Y42C
M			A26530G	D3G
			C26895T	H125Y
Orf8	C27972T	Q27stop		
	G28048T	R52I		
	A28111G	Y73C		

## Data Availability

The data presented in this study are available on request from the corresponding author.

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
