# Peer review of "VOC 202012/01 Variant Is Effectively Neutralized by Antibodies Produced by Patients Infected before Its Diffusion in Italy"

_viruses, 2021, doi:10.3390/v13020276_

Round 1
Reviewer 1 Report
This is a timely study on the neutralizing effects of the sera from patients infected with the B.1. lineage virus on the B.1.1.7 variant which first emerged in the UK. The investigators have found that the B.1. virus-elicited sera could neutralize the B.1.1.7 as effectively as B.1. lineage virus in a live virus-based neutralizing assay. The results are straightforward. It would be informative for the assessment of vaccine efficacy.
Just a few clarifications would be needed.
- The titers are quite low in neutralization against the early isolate B.1 vs. B.1.1.7 variants (1/160 ~1/320); Is that due to the sensitivity of the assay? In addition, the disease severity of the patients should be presented in the table as we know that the titer of antibodies are higher in severely ill patients.
- It would also be informative to describe in the table the B.1 defined mutations as well as those in the B.1.1.7 variants to inform the audience who have no knowledge about the exact a.a. mutations.
- Minor: use wording like VOC202012/01 instead of "English" variant as more formal nomenclature.
Author Response
Dear Editor,
Thank you for considering our manuscript and for the revisions. Below are reported the requisted revisions point by point.
Point 1: The titers are quite low in neutralization against the early isolate B.1 vs. B.1.1.7 variants (1/160 ~1/320); Is that due to the sensitivity of the assay? In addition, the disease severity of the patients should be presented in the table as we know that the titer of antibodies are higher in severely ill patients.
Response 1: We have added to Table 1 a column describing the patient's symptoms at the time of the positive test, as requested (Line 148-149).
Point 2: It would also be informative to describe in the table the B.1 defined mutations as well as those in the B.1.1.7 variants to inform the audience who have no knowledge about the exact a.a. mutations.
Response 2: We agree with you, and have decided to add a new Table (Table 2) describing the main mutations of the two lineages. (Line 173)
Point 3: Minor: use wording like VOC202012/01 instead of "English" variant as more formal nomenclature.
Response 3: Thank you for your comment. We have made the required corrections:
We replaced “English variant” with “VOC202012/01” at line 36,40,96,207
Reviewer 2 Report
This paper analyzes the efficacy of sera drawn from previously infected COVID-19 patients as a means of providing immunity against the virus. This proves to be an important alternative to the usual vaccine treatment since frequent mutations of the virus are common.
Major comment:
This paper is very well written and it addresses a crucially important topic that certainly will be of interest to health professionals across the globe. The only concern that I have is the extremely small sample used in the analysis. If these results could be replicated with a larger sample, the implications would be far-reaching and life-changing. Still, these initial results seem promising enough to warrant dissemination of this small study.
Minor comment:
Although the paper is very well written, there is some need for minor grammatical editing. For example, the last sentence of section 1.
Author Response
Dear Editor,
Thank you for considering our manuscript and for the revisions. Below are reported the requisted revisions point by point.
Major comment: This paper is very well written and it addresses a crucially important topic that certainly will be of interest to health professionals across the globe. The only concern that I have is the extremely small sample used in the analysis. If these results could be replicated with a larger sample, the implications would be far-reaching and life-changing. Still, these initial results seem promising enough to warrant dissemination of this small study.
Response: Thanks for the appropriate comment. Our group fully agrees with you, in fact we are planning to analyze a larger number of samples. However, we decided to submit the paper as soon as possible since, in our opinion, timely sharing of data, even if obtained with a limited number of samples, may be helpful to address presently challenging issues.
Minor comment: Although the paper is very well written, there is some need for minor grammatical editing. For example, the last sentence of section 1.
Response 2: Thanks for the precise report. We have made the necessary corrections:
We replaced “was” with “is” at line 73
We deleted “the” at line 75
We replaced “8” with “Eight” at line 116
We replaced “seroneutraization” with “seroneutralization” at line 131
We replaced “sera” with “serum” in Table 1
We replaced “strain” with “strains” at line 135